# Reliability of Polyetherketoneketone as Definitive Implant-Supported Bridges in the Posterior Region—An In Vitro Study of the Ultimate Fracture Load and Vertical Marginal Discrepancy after Artificial Aging

**Surakit Visuttiwattanakorn [1,\*], Apitchaya Suthamwat [2], Somchai Urapepon [3] and Sirichai Kiattavorncharoen [1]**

[1] Department of Oral and Maxillofacial Surgery, Faculty of Dentistry, Mahidol University, Bangkok 10400, Thailand
[2] Master of Science Program in Implant Dentistry, Faculty of Dentistry, Mahidol University, Bangkok 10400, Thailand
[3] Department of Prosthodontic, Faculty of Dentistry, Mahidol University, Bangkok 10400, Thailand
\* Correspondence: surakit.vis@mahidol.ac.th

**Abstract:** Purpose: This study aims to investigate the ultimate fractural strength and marginal integrity of a three-unit implant-supported fixed partial denture (FPD) framework fabricated of polyetherketoneketone (PEKK) after simulated five-year clinical aging. Materials and Methods: Twelve FPD frameworks were milled (n = 6 per group). All experimental frameworks were cemented on identical stainless-steel abutment models and subjected to five years of clinically simulated thermomechanical aging. The vertical marginal gap values were analyzed using a scanning electron microscope before and after being subjected to each aging condition. A universal testing machine was used to evaluate the ultimate fracture load. Results: A significant increase in marginal gap values of the PEKK group was observed after five years of simulation aging ($p < 0.001$), while no significant difference was seen in the titanium group ($p = 0.071$). After thermocycling, the PEKK group showed a higher statistically significant mean marginal gap value (84.99 + 44.28 μm) than before (81.75 + 44.53 μm). The titanium group exhibited superior mechanical strength, with a fracture load significantly higher than that of the PEKK group (3050 + 385.30 and 1359.14 + 205.49 N, respectively). Conclusions: Thermocycling affects the marginal gap discrepancy of PEKK restoration. However, the mean vertical marginal gap values in PEKK and titanium groups after a five-year clinical aging simulation were clinically acceptable. The ultimate fracture load values were higher than the maximum bite force reported in the posterior region. Thus, PEKK could serve as a suitable alternative material to metal in the framework of fixed dental prostheses.

**Keywords:** vertical marginal gap; polyetherketoneketone; thermocycling; cyclic loading; ultimate fracture load

## 1. Introduction

Implant-supported fixed partial dentures (FPDs) usually consist of a framework and a veneering material. The framework can be fabricated by casting metal or milling either titanium or zirconia [1]. Metal–ceramic restorations have long been used and continue to be the gold standard material for either tooth- or implant-supported FPDs [1]. Metal frameworks exhibit good mechanical properties and have predictable clinical performance [2]. Nevertheless, several drawbacks of the conventional metal–ceramic framework have been noted, such as esthetic complications due to the visible metal collar of the restoration or discoloration of gingival tissue [3], as well as suboptimal biocompatibility, which poses a potential risk of inducing allergic reactions in patients [4].

Metal-free restorations based on all-ceramic systems and polymers have been developed due to their improved esthetics and potential to overcome the shortcomings of

traditional metal-based prostheses. Polyaryletherketones (PAEKs) are a family of high-performance thermoplastic polymers with excellent properties. These materials are now well accepted and widely used in many different fields, including the automotive, semiconductor, aerospace, medical, and dental industries [5,6]. Members of this group of polymers contain ketones and ethers as functional groups that are connected to an aromatic backbone molecular chain. Two commercial PAEKs commonly used in dental applications are polyetheretherketone (PEEK) and polyetherketoneketone (PEKK) [7]. PEKK is the latest member of the PAEK family; it has a second ketone group, which improves its mechanical and physical properties [6,7], thus currently representing the apex of the quality pyramid of PEAK members.

PEKK has the properties of high dimensional stability [8], high chemical and mechanical resistance against wear, and high flexural strength. Alasdon et al. have reported that the biaxial flexural strengths of pressed and milled PEKKs are 239.9 and 257.8 MPa, respectively [9]. Therefore, PEKK has gained popularity as a framework material owing to its lightweight and manufacturing versatility, being suitable for computer-aided design (CAD)/computer-aided manufacture (CAM) milling and heat-pressed manufacturing [5,9]. Moreover, compared to PEEK, PEKK has better compatibility with different veneering materials, such as resin composites, ceramics, and acrylic resins [1].

Although the manufacturer (PEKKTON®, Cendres-Meteaux, Biel/Bienne, Switzerland) has indicated that PEKK is suitable for a wide range of prostheses due to its excellent physiological and biological properties [8], there are limited reports in the literature to date that support its clinical application [1,10]. Several studies have demonstrated that PEKK is a reliable material for long-term provisional prostheses. Elmougy stated that a monolithic provisional crown fabricated from PEKK has excellent fractural strength of 2037 N, which can withstand loads in the posterior region [8]. Another study has confirmed the ability of PEKK to restore temporary crowns and bridges; PEKK not only offers comparable results to cobalt–chrome (CoCr) restoration but also has an esthetic advantage [10]. Han et al. reported on a case of an implant-fixed prosthetic restoration with a framework fabricated using PEKKTON® in a patient with a fully edentulous maxilla and partially edentulous mandible, though they also considered it suitable for long-term provisional restoration [1]. Thus, there are doubts as to whether the PEKK framework is durable enough for a definitive implant-supported fixed partial prosthesis.

A previous study reported that the fracture load resistance of PEEK-made three-unit FDPs is 1383 N, indicating that PEEK might be an appropriate alternative material for FDPs, especially in load-bearing areas [11]. In 2016, Nazari reported similar results in a study that examined the fracture strength of three-unit implant-supported FDPs with excessive crown height fabricated from different materials. The mean fracture loads for Ni–Cr, zirconia, and PEEK prostheses were 5591, 2086, and 1430 N, respectively. The authors concluded that although the PEEK framework presented the lowest structural strength, it was still capable of withstanding bite force in the molar region, thus being sufficient for clinical application [12]. The fracture strength of three-unit posterior unveneered bridges made of PEKK presented a value greater than 1100 N; on the other hand, veneered bridges had a value greater than 2500 N. However, as these results were presented by the manufacturer, further laboratory and clinical studies are required to determine whether this framework is suitable for clinical use.

Marginal adaptation is another important factor that determines the longevity of dental restorations. A misfit between the framework prosthetic and implant platform or abutment could result in improper stress distribution or a higher amount of stress on the bone-implant interface and underlying structures, leading to mechanical and/or biological complications [13]. As has been reported by many authors, polymer base materials may show creep deformation related to temperature, time, and loading due to their mechanical properties [14]. Applying PEKK as a structural material may lead to deformation under long-term loading and compromise the fit of the framework to the implant.

Clinical studies are typically time-consuming and expensive, and the standardization of the test parameters is difficult. However, static fracture strength tests in isolation do not represent a clinical situation, as the restoration service in an oral cavity is generally subjected to dynamic, repetitive loading. Therefore, in vitro dynamic loading tests allow for the evaluation of dental restorative systems under clinically relevant conditions while saving evaluation time.

The purpose of this in vitro study was to investigate the fractural strength of three-unit fixed partial prosthesis frameworks fabricated from PEKK as well as the marginal integrity of the PEKK framework after a simulation of five years of clinical service. The null hypothesis is that PEKK is suitable for use as a framework material with comparable durability to the metal-based gold standard; thus, mechanical loading would not affect the marginal fit of a restoration formed from PEKK.

## 2. Materials and Methods

A brief experimental workflow of this study is illustrated in Figure 1.

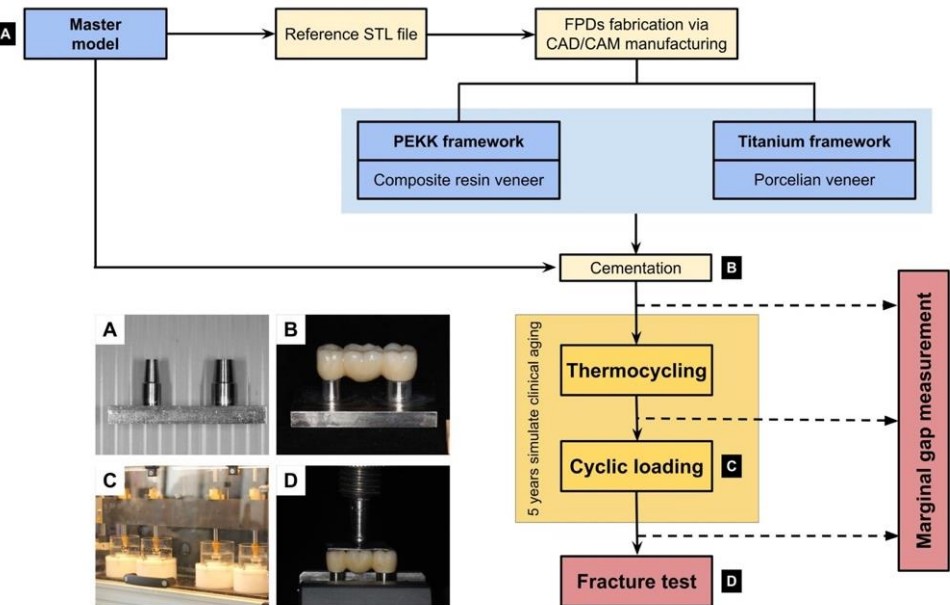

**Figure 1.** A brief experimental workflow of this study. The master models (**A**) were scanned, and a CAD/CAM manufacturing system was used for the fabrication of the PEKK and titanium frameworks. The FPDs were cemented to the models (**B**), then underwent simultaneous thermal cycling and mechanical loading (**C**) procedures to simulate 5 years of clinical service. Vertical gap measurements were performed before and after each artificial aging condition. Fracture load after aging was measured (**D**).

### 2.1. Model Construction

In this in vitro study, two implant abutments (Cementable abutment, Straumann, Basel, Switzerland), with a height of 5.5 mm and a 5 mm and 6.5 mm platform, were used as the premolar and molar, respectively. The abutments were scanned (3ShapeD810; 3Shape, Copenhagen, Denmark), and the master model was designed with the dimensions shown in Figure 2. The distance between the centers of the abutments was 16.5 mm, approximately equal to the distance between the centers of the mandibular second premolar and the second molar. Then, the entire model was scanned to fabricate 12 identical stainless-steel abutment models with a base diameter of 30 mm × 15 mm × 5 mm using a milling machine (Milstar MVL 850/M80 Control; Mitsubishi, Tokyo, Japan).

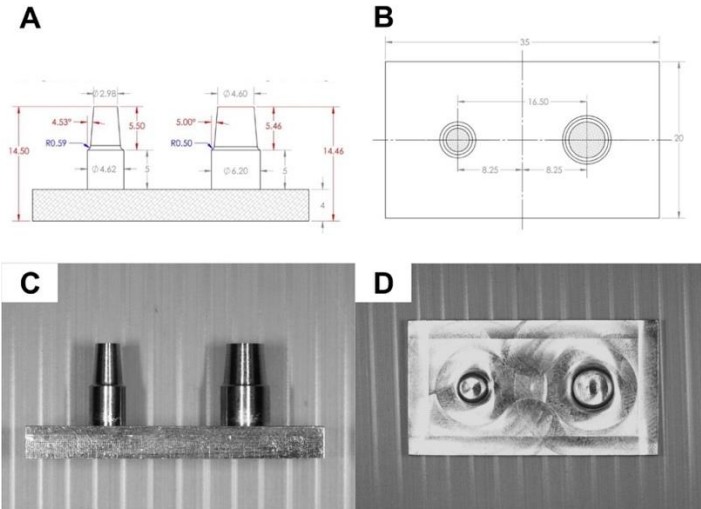

**Figure 2.** (**A**,**B**) Illustrations of CAD design and of an abutment model. (**C**,**D**) Photographs of a milled abutment model.

### 2.2. Framework Fabrication

The appropriate sample size was calculated using a one-way ANOVA with a power of 90%. The minimum sample size was calculated as four per group; however, the sample size was increased to six per group to increase the significance level of this study.

Two groups of mandibular posterior three-unit FPDs were constructed. The control group was manufactured from titanium alloy (CORITEC TI DISC TITANIUM GRADE 5, IMES-ICORE GMBH, Eiterfeld, Germany), while the other group was fabricated from PEKK (PEKKTON®, Cendres-Meteaux, Biel/Bienne, Switzerland).

The CAD/CAM manufacturing system was used for the fabrication of the PEKK frameworks. The reference models were scanned using a 3ShapeD810 scanner (3ShapeD810; 3Shape, Copenhagen, Denmark). Using CAD software, the substructures were designed in a supported anatomical form with a circular framework thickness of 0.6 mm and 0.8 mm in the occlusal plane. The connectors had a cross-sectional area of 16 mm$^2$ [2], an occluso-gingival height of 4.45 mm, and a buccolingual width of 3.6 mm [15]. The STL data were then transferred to the CAM unit (CORiTEC450i; Imes-icore GmbH, Eiterfeld, Germany), where the PEKK blocks were milled.

The same process was used to fabricate the titanium frameworks. The circular framework had a thickness of 0.6 mm and 0.8 mm in the occlusal plane. The connector's dimensions were 3 mm in height and 2.5 mm in width, with a cross-sectional area of 7.5 mm$^2$ [2]. A CAM unit (DMG Ultrasonic 10, ULTRASONIC Series, DMG MORI, Tokyo, Japan) was used to mill the frameworks from the titanium disks according to the manufacturer's instructions.

### 2.3. Veneering of the Frameworks

To standardize the veneering contour on all specimens, one of the titanium frameworks was veneered with VITA TITANKERAMIK (VITA Zahnfabrik, Bad Säckingen, Germany), and the silicone index was fabricated. For the PEKK group, all frameworks were blasted with 110 μm aluminum oxide particles at 2–3 bar pressure, followed by the application of an adhesive Vasio.link (Bredent GmbH & Co. KG, Senden, Germany) and light curing for 90 s. The PEKK frameworks were then veneered with VITAVM®LC (VITA Zahnfabrik, Bad Säckingen, Germany).

All frameworks were randomly cemented on the steel abutment models using a resin composite cement, Multilink® Speed (Ivoclar Vivadent, Schaan, Liechtenstein), in standard fashion [15,16] and according to the manufacturer's instructions. The restorations were placed on the models, and excessive cement was removed. A special cementation device

was used to ensure correct cement distribution by applying a compressive force of 50 N for 10 min. All specimens were then stored in water (37 °C) for 24 h.

### 2.4. Artificial Aging (Thermal Mechanical Cyclic Loading: TMCL)

All testing frameworks (n = 6 per group) underwent simultaneous thermal cycling and mechanical loading procedures to simulate five years of clinical service [17,18]. Before being subjected to mechanical loading, the specimens were thermally aged in a thermocycler (TC 400; King Mongkut's Institute of Technology Ladkrabang, Bangkok, Thailand) for 12,000 cycles in distilled water at temperatures of 5 and 55 °C with a dwell time of 30 s [19].

After thermal aging, each model was mounted on an oral simulator (CS-4.4; Professional, Germany) to perform fatigue loading using 1,200,000 mechanical loads at a frequency of 1.6 Hz with a load of 50 N applied to the middle of the pontic site.

### 2.5. Marginal Discrepancy

To examine the discrepancy between frameworks, the marginal gap was analyzed using a scanning electron microscope (SEM; JSM-6610LV; JEOL Ltd., Tokyo, Japan) before and after each artificial aging condition. All measurements were performed by a trained investigator. A total of seven reference points were marked on each abutment to guide the measurement of the vertical gap between the framework and the abutment platform (Figure 3). These measuring points were marked using an indelible marking pen, assisted by a customized acrylic template, and served as reference points for the investigator [20]. The data were captured and digitized using the software. To increase the number of measurements per specimen, another 29 lines parallel to the reference line were created, with each line 10 μm apart. (Figure 4) Each line was measured using the software's caliper, and the average mean was calculated. Thus, seven different points with 210 measurements were taken on one abutment [20].

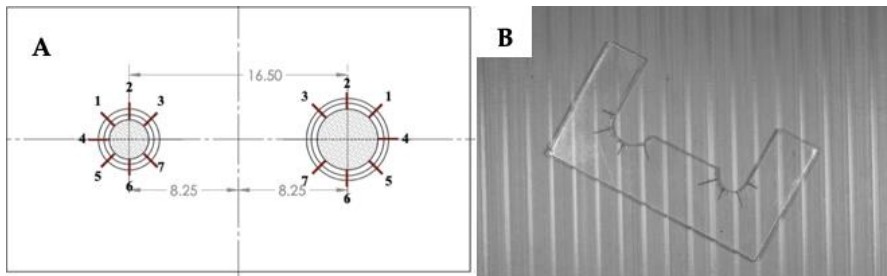

**Figure 3.** (**A**) Illustration of seven reference points on abutment model. (**B**) Photographs of a customized acrylic template.

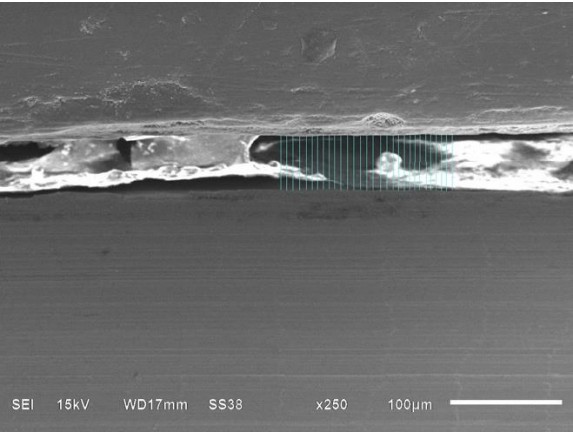

**Figure 4.** Vertical marginal gap measurement using a scanning electron microscope (SEM) at ×250 magnification.

### 2.6. Fracture Resistance

After aging and examining the marginal discrepancy, specimens that survived the simulated-age fatigue were loaded onto a universal testing machine (Instron 5566; Instron Ltd., Buckinghamshire, England) for examination of their fracture strengths. Force was applied using a steel ball (diameter: 5 mm) with 1 mm of tin foil placed between the frameworks and an antagonist metal ball to achieve even force distribution and prevent force peaks. Force was applied to the middle of the pontic at a crosshead speed of 1 mm/min. Application of the load was continued until the fracture load dropped to more than 15% of the maximum load.

### 2.7. Statistical Analysis

Statistical analysis was performed using SPSS software (version 18.0; IBM Corp., Armonk, NY, USA). The data were explored for normality by checking the data distribution using Shapiro–Wilk testing. The mean values and standard deviations of the vertical marginal gap discrepancies were calculated for each group and measurement point. The Wilcoxon signed-rank test and paired-sample *t*-test were conducted to compare the marginal fit before and after each aging condition for the titanium and PEKK groups, respectively.

A student's *t*-test was carried out to compare the mean values and standard deviations of the marginal gap change between the groups. The Friedman test was used to detect statistical differences in the mean marginal gap change between the measurement points. A student's *t*-test was also performed to compare the mean fracture load between the test groups. The significance level was set at $p = 0.05$.

## 3. Results

The overall mean marginal gap values and standard deviations (SDs) for the experimental groups are displayed in Table 1. The paired-sample *t*-test results indicate a significant increase in the marginal gap values of the PEKK group after five years of simulated aging ($p < 0.001$). No significant difference was observed in the titanium group ($p = 0.071$). After thermocycling, it was found that the PEKK group showed a significantly higher mean marginal gap value (84.99 ± 44.28 μm) than before (81.75 ± 44.53 μm). In the titanium group, however, no significant difference was found when comparing before (44.04 ± 24.16 μm) and after (44.65 ± 24.15 μm) thermocycling. There was no significant influence of cyclic loading on the vertical marginal gap in either experimental group (Table 1).

**Table 1.** Mean (μm) values of vertical marginal gap of three-unit FPDs initially, after different aging conditions and a five-year simulation.

| Group (n) | Mean Marginal Gap (SD) | | | | | | |
|---|---|---|---|---|---|---|---|
| | Initial | After Thermocycling | *p*-Value | After Cyclic Loading | *p*-Value | After Five-Year Thermomechanical Aging | *p*-Value |
| Titanium (12) Percentile: 25/75 | 44.04 (24.16) 30.12/59.32 | 44.65 (24.15) 29.15/58.24 | 0.209 | 44.72 (24.11) 28.61/60.83 | 1.00 | 44.72 28.61/60.83 | 0.071 |
| PEKK (12) Percentile: 25/75 | 81.75 (44.53) 36.36/120.50 | 84.99 (44.28) 39.11/121.65 | 0.000 [a] | 85.31 (44.43) 39.31/123.85 | 0.323 | 85.31 39.31/123.85 | 0.000 [a] |

[a] indicates statistically significant difference (*p*-value < 0.05).

The means and SDs of the change in marginal gap values for the tested groups after each aging condition are shown in Table 2. Significant differences were observed in post-thermal cycling and post-simulated aging among the groups. The Student's *t*-test results indicate that the PEKK group had a significantly higher marginal gap change than the titanium group after thermocycling ($p = 0.003$) and five-year simulated aging ($p = 0.002$). When differences in marginal gap change values between measuring points (Figure 3) were analyzed within each group, the Friedman test results revealed no statistically significant differences (Table 3).

**Table 2.** Mean (μm) values of marginal gap change after different aging conditions.

| Group (n) | Mean of Marginal Gap Change (SD) | | |
|---|---|---|---|
| | Post-Thermocycling | Post-Cyclic Loading | Post-Five-Year Thermomechanical Aging |
| Titanium (12) | 0.04 (2.07) | 0.93 (2.71) | 0.71 (1.23) |
| PEKK (12) | 3.24 (2.01) | 0.32 (1.07) | 3.56 (2.02) |
| *p*-value | 0.003 [a] | 0.45 | 0.002 [a] |

[a] indicates statistically significant difference (*p*-value < 0.05).

**Table 3.** Mean (μm) values of marginal gap change after different aging conditions according to positions.

| Group | Position | Mean Marginal Gap Change (Δ) According to Position (μm) | | |
|---|---|---|---|---|
| | | ΔPost Thermocycling | ΔPost Cyclic loading | ΔPost Five-Year Aging |
| Titanium | 1 | −0.45 | 1.90 | 1.45 |
| | 2 | 0.61 | 1.38 | 1.99 |
| | 3 | 1.49 | −0.84 | 0.65 |
| | 4 | 0.85 | −0.03 | 0.33 |
| | 5 | −0.63 | 1.53 | 2.75 |
| | 6 | −0.53 | 1.34 | 0.81 |
| | 7 | 0.50 | −0.80 | −0.70 |
| | *p*-value | 0.679 | 0.829 | 0.631 |
| PEKK | 1 | 3.17 | 0.25 | 3.42 |
| | 2 | 2.18 | 0.97 | 3.15 |
| | 3 | 2.04 | 0.19 | 2.22 |
| | 4 | 3.77 | 0.65 | 4.41 |
| | 5 | 2.36 | 0.84 | 3.20 |
| | 6 | 4.46 | −0.80 | 3.66 |
| | 7 | 5.11 | 0.12 | 5.23 |
| | *p*-value | 0.514 | 0.532 | 0.532 |

The fracture load results are presented in Table 4. A statistically significant difference was observed between the two groups (*p* < 0.001), where the titanium group exhibited superior mechanical strength with a fracture load higher than that of the PEKK group (3050 ± 385.30 and 1359.14 ± 205.49 N, respectively). Figure 5 provides a graph of mean load (N) vs. extension (mm) for both experimental groups.

**Table 4.** Fracture load (N) of three-unit FPDs fabricated with different materials after five-year simulated aging.

| | Ultimate Fracture Load (N) | | *p*-Value |
|---|---|---|---|
| | Titanium | PEKK | |
| Mean | 3050.18 | 1359.14 | *0.000 [a]* |
| (SD) | (385.30) | (205.49) | |

[a] indicates statistically significant. (*p*-value < 0.05).

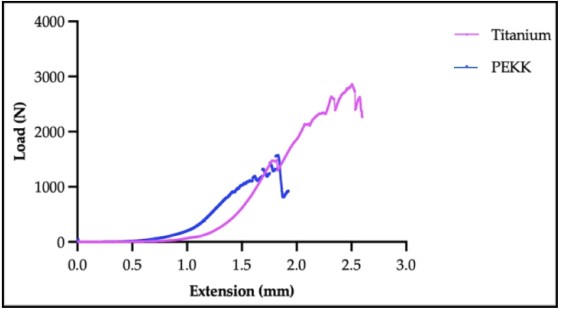

**Figure 5.** Load-extension graphs of experimentally veneered FPDs fabricated with PEKK and titanium group.

## 4. Discussion

We investigated and compared the marginal fit and fracture resistance of restorations depending on whether they were manufactured from the latest high-performance polymer, PEKK, vs. conventional metal–ceramic. All examined FPDs survived five years of simulated clinical aging without prosthesis failure. The results support the rejection of the null hypothesis, as significant differences were observed in the comparison of the vertical marginal gap change after aging and the ultimate fracture strengths of the groups.

The marginal fit of a restoration is considered one of the most important criteria determining the success of fixed dental restorations [20]. A consensus on the maximum clinically acceptable marginal discrepancy has not been developed; however, the most widely used value in the literature is 120 μm [20,21], proposed by McLean and von Fraunhofer [22]. The authors found that it is difficult to detect marginal gap values of <80 μm using exploratory or direct tactile stimuli. With a blunt instrument, these values reached 160 μm; however, gap values > 160 μm could be seen in the X-ray, which is considered to indicate a poor clinical fit restoration. Thus, a restoration achieving a marginal gap of <120 μm ensures a proper clinical fit [22]. However, maximum tolerable marginal discrepancies ranging from 50 to 200 μm have been reported [23,24]. In the present study, no specimen showed a mean marginal gap value greater than 120 μm, further indicating that the PEKK framework potentially meets the requirements for successful restoration in clinical practice. Our results are consistent with those of many other studies, which have shown that all marginal gaps of PEKK or PEEK restorations ranged from 20 to 90 μm, indicating they are clinically acceptable [25–27].

Dental restorations are subject to changes in moisture, dynamic temperatures, and load occurring within the oral cavity. In vitro experiments have shown that thermomechanical aging is essential for investigating material behavior under clinical-like conditions [28–30]. Thermomechanical testing may have various effects on the dimensional stability, mechanical properties, and marginal integrity of a restoration. For polymers, the effects may include polymerization stresses, water sorption, plasticizing effects, hygroscopic expansion, and cyclic deformation [8,29–32].

The results obtained after a five-year aging simulation suggest that PEKK and titanium, as framework materials, may perform differently in clinical use over time. Titanium frameworks can withstand thermomechanical aging and demonstrate excellent marginal fit stability. On the other hand, a significant increase in the marginal discrepancy was observed in the PEKK group ($p < 0.001$). The results reveal a negative impact of thermocycling on marginal fit only in the PEKK group; however, cyclic loading had no influence on the marginal integrity of either experimental material.

When dental prostheses undergo various temperature changes, they face dynamic expansion and contraction, especially at the thin margin area, which may cause material distortion and increase the marginal gap [29,31]. The results of early studies showed that the margin fit of resin-based polymer restorations changes after thermal cycling [30–32]. Ehrenberg et al. (2000) observed a significant increase in the marginal discrepancy after thermal cycling of two conventional polymethyl methacrylate (PMMA) and bis-acryl resin temporary crowns, with mean changes in the marginal gap size of 43.9–548.9 μm. The mean change in the marginal discrepancy of the PEKK framework (3.24 μm) in this study was considerably lower than the values reported for other resin-based polymers in earlier studies. According to the temperature performance of thermoplastic polymers, they soften when heated to the glass transition temperature (Tg), changing from a brittle to a more ductile form [33]. The Tg of PEKK is higher than that of PMMA (163 and 106 °C, respectively) [34]. The temperature used in the aging protocol was 55 °C, closer to the PMMA than the PEKK Tg. Therefore, PEKK may have higher marginal fit stability than PMMA.

In our study, thermocycling did not affect the marginal fitness of the titanium framework ($p = 0.029$). Titanium presented only a small increase in the mean marginal gap size (0.4 μm), whereas PEKK exhibited an almost five-fold increase (3.24 μm). The reason for these findings may be the different water absorption abilities and thermal expansion rates of

the two materials. Thermal expansion is the tendency of substances to change their volume, shape, and area in response to a change in temperature. The degree of expansion divided by the change in temperature is the coefficient of thermal expansion (CTE), where lower CTE values indicate a lower tendency to change in size. In general, solid materials with higher melting points are more likely to have a lower thermal expansion due to their higher bond energy [35]. Therefore, titanium may exhibit more favorable dimensional stability than PEKK, due to its extremely high melting temperature (Tm) of 1660 °C and lower CTE of $10.3 \times 10^{-6}$ C$^{-1}$, according to the manufacturer's data. For the PEKK material, a melting temperature of 363 °C and a CTE of $47 \times 10^{-6}$ C$^{-1}$ have been reported [33].

The absorption of water within a material may cause dimensional changes. The volumetric expansion of resin-based polymers has been reported to range between 0.7 and 1% [36]. A study investigating the hygroscopic expansion change of PEKK (PEKKTON®) revealed that PEKK (0.14% ± 0.14%) specimens had the lowest percent volume change over the entire study in comparisons with Vita Enamic (0.38% ± 0.16%) and Lava Ultimate (1.06% ± 0.14%) [8]. A similar study has reported that PEEK showed the lowest solubility and water absorption values among different CAD/CAM polymers, including PMMA [37]. Although the marginal fit of PEKK restorations can be affected by water absorption, the values of discrepancy would be minimal compared with those of other dental polymers or composite materials.

In our study, cyclic loading was not found to affect the marginal integrity of the tested restorations. Thus, the assumption of marginal gap discrepancy enlargement as a result of framework bending from plastic deformation or creep under fatigue loading of ductile material is rejected. This may be partially due to the very high impact strength at low temperatures, high mechanical fatigue strength, and very low tendency to creep in both experimental materials [38]. A study evaluating the long-term mechanical properties of PEEK has indicated a low creep strain of less than 0.1% after 2000 h of loading, suggesting that the nonrecovery deformation of PEEK would be negligible in clinical practice [39]. Moreover, the load value used in our study was considerably lower than the elastic limits of both materials.

Our study was performed on cemented FDPs, and our intention was to simulate actual clinical situations. However, cementation can increase marginal gaps [40] and may affect marginal adaptation. Various factors could potentially affect the cement gap, including cement viscosity, the pressure of cementation, time of pressure application, and degree of preparation taper [41,42]. Therefore, the cementation of specimens in this in vitro study was carried out using a loading cementation device with a load of 5 kg in order to ensure full seating [42,43].

No general guidelines or gold standards exist regarding methods that measure the marginal adaptation of FDPs. Various techniques have been proposed to analyze marginal precision, but no conclusive evidence indicates that a single method is superior to others [20,44]. The direct-view technique is the most commonly used, followed by the cross-sectional method and the impression replica technique [44]. In this in vitro study, the direct viewing technique using SEM was chosen because it is noninvasive, less time-consuming, and easy to repeat. Most importantly, it allowed for the evaluation of simulated clinical experiments. To achieve clinically relevant information and a consistent estimate of the precision of fit, the literature suggests that at least 50 measurements on a specimen should be collected [45]. In addition, data obtained from measuring the experimental crown margin at fewer than 18 points might be misleading [46]. Thus, the number of measurements in this study was 210 per specimen to ensure accurate results, and the same operator performed all measurements.

In the present study, the ultimate fracture strength was also evaluated, as this is another important factor in predicting the clinical service and failure rates of the prosthesis material. The titanium FDPs fractured at a mean load of 3018 ± 385.3 N. This value is similar to a previous study reporting a fracture load of 3500 N for three-unit metal–ceramic FPDs [47]. The PEKK frameworks showed a significantly lower mean fracture load of 1359.14 ± 205.49 N as a result of lower compressive and tensile strengths than the

titanium group. Several previous in vitro studies have reported a mean fracture load in the range of 1070–2350 N for PEKK and PEEK FPDs [11,15,16,48]. Experimental designs and testing parameters, such as the dimensions of the framework connector, cementation, abutment material, fabrication technique, veneering material, applied load, and aging conditions, differ from previous in vitro studies. Thus, it is difficult to compare the results of these studies with those obtained here. However, the value presented in this study is still higher than the maximum bite force reported in the posterior region, ranging from 600 to 920 N [49,50]. This indicates the potential for clinical use of PEKK in implant-supported FPD frameworks in the molar region.

There are some limitations to the present study. All FDPs were fabricated under standardized conditions, which may not reflect the conditions in clinical practice. The fit and fracture strength of the restoration may be influenced by numerous factors, such as the fabrication technique, milling machine, and cementation [11,15,16]. Further studies are needed to analyze the factors that affect the marginal gap of PEKK restorations, including the technique of fabrication, different milling machine axes, and the luting agent used. In addition, further investigations with larger samples and clinical trials are needed to validate the obtained results.

**5. Conclusions**

Within the limitations of this in vitro study, the following conclusions are drawn:

1. The vertical marginal gap of both the PEKK and titanium groups exhibits a clinically acceptable marginal fit of less than 120 μm.
2. The thermal aging process affects the marginal fit of PEKK but not that of titanium.
3. The fracture load test indicates the potential for clinical use of PEKK in an implant-supported FPD framework in the molar region.

**Author Contributions:** Conceptualization, S.V. and A.S.; methodology, S.U. and A.S.; validation, S.V. and S.U.; formal analysis, A.S.; investigation, A.S.; writing—original draft preparation, A.S.; writing—review and editing, S.V.; supervision, S.V., S.U. and S.K.; project administration, S.V. and S.K. All authors have read and agreed to the published version of the manuscript.

**Funding:** This research received no external funding.

**Institutional Review Board Statement:** Not applicable.

**Informed Consent Statement:** Not applicable.

**Conflicts of Interest:** The authors declare no conflict of interest.

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
