# Peer review of "Reliability of Polyetherketoneketone as Definitive Implant-Supported Bridges in the Posterior Region—An In Vitro Study of the Ultimate Fracture Load and Vertical Marginal Discrepancy after Artificial Aging"

_applsci, doi:10.3390/app122211454_

Round 1

Reviewer 1 Report

This study investigated the marginal fit and fracture load of the PEKK bridges by comparing the titanium bridges. The results indicated that the PEKK bridges can be acceptable to use in the framework of fixed dental prostheses. The outcome contributes to dentistry from the viewpoint of fundamental research. I recommend this paper to the journal after some revisions. The comments are listed as follows.

1.      Observation positions for the marginal gap assessment.

Please clarify the observation positions for the marginal gap by SEM. I recommend new figure to indicate them.

2.      Tables 1, 2 and Figure 4.

The results for the marginal gap are duplicated in each table and figure. These tables and figure should be put together in one table or figure.

3.      Table 4

We cannot the observation position for the SEM. As mentioned our comment 1, It should be clear where the observation point is. 

4.      Table 5

Table 5 contains every data for the sample. However, we cannot understand why every data is needed. I recommend only mean values (and SD) are indicated in the table.

5.      Figure 5

Why is every graph given in the figure? What do you mean by this result? Is every graph needed? If no, representative ones for PEKK and Ti should be compared in one graph.

6.      Novelty

There is adequate discussion. However, we cannot the novelty of the present work. Please clarify it.

Reviewer 2 Report

The manuscript entitled „ Reliability of Polyetherketoneketone as Definitive Implant-Supported Bridges in the Posterior Region—An In Vitro Study of the Ultimate Fracture Load and Vertical Marginal Discrepancy after Artificial Aging”, submitted for evaluation to Applied Sciences, concerns the investigation of utility of polymer PEEK for long-term dental restoration during in vitro tests. The results clearly show that PEKK could serve as a suitable alternative material to metal in the framework of fixed dental prostheses..

            This topic is interesting and final observation useful in terms of design the dental therapeutic model.

The work is nice and clearly written in the correct language. The description of the methodology is precise and the conclusions are appropriate. I only noticed minor errors. My comments and questions concerning the submitted article are listed below:

 COMMENTS TO AUTHORS

 1. Please check the value 16 mm in line 148. I suspect that it should be in squared mm.

2. Figure 4 is not cited in the text.

Reviewer 3 Report

Manuscript No: applsci-1990046

Authors: Surakit Visuttiwattanakorn, Apitchaya Suthamwat, Somchai Urapepon and Sirichai Kiattavorncharoen

Article title: Reliability of polyetherketoneketone as definitive implant-supported bridges in the posterior region - an in vitro study of the ultimate fracture load and vertical marginal discrepancy after artificial aging

This article discusses the issue of fractural strength and marginal integrity of polyetherketoneketone frameworks, used for fabrication of implant-supported fixed partial dentures. The text of the article follows the generally accepted rules and includes all necessary sections: introduction, materials and methods, results, discussion and conclusions.

The introduction provides sufficient information about the topic discussed. At the end of this part, the aim of the study was clearly formulated, and the null hypothesis was declared. The experimental workflow of the study is described in detail, and it is illustrated with three figures. To compare the marginal fit of the implant-supported FPD’s the following statistical methods are used: Shapiro–Wilk test, Wilcoxon signed-rank test and paired-sample t-test. The results are clearly presented in five tables and two graphs. In the discussion section, the results obtained are compared to the similar ones conducted in previous studies. All conclusions are supported by the results. All the cited references are relevant to the research.

Minor spell check required and the exact places are indicated in the attached file. As a result of the arguments above, I believe that the article can be published in its present form.

Author Response

Dear Reviewer3 

I have been edited following all your comments. 

Best regards